# AutoHete: An Automatic and Efficient Heterogeneous Training System for LLMs

## Abstract

Transformer-based large language models (LLMs) have demonstrated exceptional capabilities in sequence modeling and text generation, with improvements scaling proportionally with model size. However, the limitations of GPU memory have restricted LLM training accessibility for many researchers. Existing heterogeneous training methods significantly expand the scale of trainable models but introduce substantial communication overheads and CPU workloads. In this work, we propose AutoHete, an automatic and efficient heterogeneous training system compatible with both single-GPU and multi-GPU environments. AutoHete dynamically adjusts activation checkpointing, parameter offloading, and optimizer offloading based on the specific hardware configuration and LLM training needs. Additionally, we design a priority-based scheduling mechanism that maximizes the overlap between operations across training iterations, enhancing throughput. Compared to state-of-the-art heterogeneous training systems, AutoHete delivers a 1.32x~1.91x throughput improvement across various model sizes and training configurations.

## 1 Introduction

The Transformer architecture (Vaswani et al., 2017) has become the foundation of Large Language Models (LLMs) due to its excellent performance across various natural language processing (NLP) tasks (Raffel et al., 2020; Devlin et al., 2018; Yang et al., 2019). Scaling laws (Kaplan et al., 2020) suggests that model quality increases in model size, leading to the development of ever-larger models like OPT (Zhang et al., 2022), BLOOM (Le Scao et al., 2023), and the GPT series (Radford et al., 2018; 2019; Brown et al., 2020; Achiam et al., 2023). However, GPU memory capacity has increased only 2.5-fold, from NVIDIA's 32 GB V100 to the 80 GB H100, a minor increase compared to the thousand-fold growth in model sizes over the past five years. This "GPU memory wall" (Gholami et al., 2024) restricts the accessibility of LLM training for most researchers. Efforts to tackle this issue mainly involve parallelism and memory-saving strategies for LLM training.

Leveraging data (Li et al., 2020), model (Shoeybi et al., 2019), pipeline (Huang et al., 2019), and hybrid parallelism (Zheng et al., 2022), we can distribute activations, parameters, and optimizer states across multiple GPUs, enabling scalable LLM training with improved efficiency. However, this requires many GPUs, often too costly for most academic teams and small businesses, as training a 10-billion parameter model needs 16 NVIDIA V100 GPUs (Ren et al., 2021). Conversely, memory-saving techniques like activation checkpointing (Kirisame et al., 2020; Zhao et al., 2023) and heterogeneous training (Ren et al., 2021; Fang et al., 2022; Sun et al., 2022) allow significant scaling on a single GPU.

To democratize transformer-based LLM training, we focus on memory-saving approaches that complement distributed training methods to enhance model scalability and speed up training. Although activation checkpointing reduces GPU memory usage by discarding and recomputing activation tensors, it fails to adequately address the extensive memory needs for model data, which include parameters, gradients, and optimizer states that are crucial for LLM training. Heterogeneous training presents a promising solution for LLMs by offloading model data to cheaper CPU memory. ZeRO-Offload (Ren et al., 2021) enables training of models 10x larger on a single NVIDIA V100 GPU by offloading all optimizer states to the CPU, albeit at the cost of significant CPU workload and communication overhead. PatrickStar (Fang et al., 2022) and StrongHold (Sun et al., 2022) dynamically allocate model data across heterogeneous memory spaces, achieving a better trade-off. However, ex-

isting heterogeneous training systems exhibit two key limitations. First, they fail to comprehensively analyze the memory requirements and execution costs of all data involved in training, including activations, parameters, gradients, and optimizer states. Second, significant resource idle periods arise between consecutive training iterations, caused by lagged gradient offloading and CPU optimizer updates. Prior works limit operations overlap to individual iterations, ignoring opportunities for overlap across iterations.

This paper introduces AutoHete, an automatic and efficient heterogeneous training system that dynamically integrates activation checkpointing, parameter offloading, and optimizer offloading. Achieving both flexibility and efficiency poses two challenges for the system: (1) Given a specific LLM architecture, batch size, and GPU-CPU configuration, the system must decide which intermediate activations to recompute, which parameters to offload, and which optimizer states to place on the CPU. (2) Another challenge lies in scheduling operations across different execution streams—including GPU computation, offloading, prefetching, and CPU optimizer updates—to maximize the overlap while preserving data dependencies.

To tackle these challenges, AutoHete involves a two-stage optimization strategy. Firstly, we construct a cost model to accurately capture GPU peak memory consumption and execution time during heterogeneous training. We then formalize the heterogeneous training optimization problem as an integer linear program (ILP), deriving the optimal plan in a few seconds. Instead of uniform post-backward parameter updates, our cost model considers asynchronous layer-wise optimizer steps during the backward pass, opening up more granular scheduling opportunities for optimal operation overlap. The second stage introduces a priority-based scheduling strategy to minimize resource idle periods between successive training iterations. By prioritizing gradient offloading and CPU optimizer updates for earlier LLM layers, it facilitates the early initiation of subsequent training iterations. This work makes the following key contributions:

- We formulate the heterogeneous training optimization problem as an ILP problem, which combines activation checkpointing, parameter offloading, and optimizer offloading.
- We introduce a priority-based scheduling strategy, which achieves operators overlapping across training iterations without increasing peak GPU memory usage.
- We evaluate AutoHete's efficiency and adaptability in both single-GPU and multi-GPU environments, demonstrating a 1.32x~1.91x performance improvement over state-of-the-art heterogeneous training solutions.

## 2 BACKGROUND AND RELATED WORK

### 2.1 TRANSFOMER-BASED LLM TRAINING

LLMs like OPT (Zhang et al., 2022) and GPT series (Radford et al., 2018; 2019; Brown et al., 2020; Achiam et al., 2023) stack multiple transformer blocks for powerful sequence modeling and representation learning. While pre-training requires vast computational resources dominated by tech companies, fine-tuning on domain-specific tasks (Touvron et al., 2023) enables broader participation but still faces significant memory challenges.

GPU memory during training is consumed by activations and model data (parameters, gradients, optimizer states). Using mixed-precision training with ADAM optimizer (Ren et al., 2021), activations and parameters are stored in FP16 while maintaining FP32 precision for optimizer states, which include momentum, variance, and FP32 parameter copies. This requires $16M$ bytes for a model with $M$ parameters, meaning a 2-billion parameter model would exceed a single V100 GPU's memory capacity even before accounting for activations, which scale with batch size and sequence length. Memory fragmentation further constrains effective utilization, making these resources extremely scarce. In response, research efforts are increasingly focused on enhancing LLM training through multi-GPU parallelism and single-GPU memory optimization techniques.

### 2.2 PARALLEL TRAINING

**Data Parallelism (DP)** (Li et al., 2020; Jiang et al., 2020; Peng et al., 2019; Renggli et al., 2019) distributes inputs across GPUs while maintaining full model copies. However, DP alone cannot

handle LLM training due to single-GPU memory constraints. ZeRO (Rajbhandari et al., 2020) addresses this by partitioning model states across GPUs, using efficient reduce-scatter and all-gather operations to minimize communication overhead.

**Model Parallelism (MP)** (Shoeybi et al., 2019; Zeng et al., 2022) divides model parameter across GPUs. Mesh-Tensorflow (Shazeer et al., 2018) enables flexible tensor partitioning, while Megatron-LM (Shoeybi et al., 2019) optimizes communication through strategic row and column parallelism for transformer-based LLMs.

**Pipeline Parallelism (PP)** (Huang et al., 2019; Narayanan et al., 2019) segments models into stages, processing micro-batches in a pipelined manner. Efficient scheduling strategies (Li & Hoefler, 2021; Sun et al., 2024) are crucial for balanced GPU utilization and throughput.

**Hybrid Parallelism** (Zheng et al., 2022; Yuan et al., 2024; Jiang et al., 2024), combining DP, MP, and PP, has also been developed to achieve optimal distributed training efficiency. Nevertheless, these parallelism solutions necessitate sufficient aggregated GPU memory to accommodate model data, presenting a significant affordability challenge.

### 2.3 MEMORY-SAVING TRAINING

**Activation checkpointing** reduces memory usage by retaining only essential activations as checkpoints while recomputing discarded activations from these checkpoints during the backward pass. A typical approach segments an $N$-layer model into $\sqrt{N}$ parts (Chen et al., 2016). Various strategies have been proposed, from optimal but computationally intensive solutions (Jain et al., 2020) to faster but sub-optimal approaches (Kirisame et al., 2020; Zhao et al., 2023), each balancing memory savings against recomputation costs.

**Offloading** leverages CPU memory to supplement GPU memory during training. While initially developed for CNNs to manage activations (Rhu et al., 2016), LLM training shifts focus to handling model data. L2L (Pudipeddi et al., 2020) pioneered heterogeneous LLM training by keeping only one transformer layer on GPU. ZeRO-Offload (Ren et al., 2021) enables training of 13-billion parameter models on a single V100 GPU by offloading gradients and optimizer states to CPU, though its static approach limits efficiency. PatrickStar (Fang et al., 2022; Li et al., 2023) and StrongHold (Sun et al., 2022) introduced dynamic memory allocation methods for model data using synchronous and asynchronous execution, respectively. However, the absence of comprehensive analyses of memory usage and execution overhead during training results in suboptimal performance.

This paper focuses on GPU memory-saving training to make LLM training more accessible. Existing approaches lack flexibility and do not jointly consider activation checkpointing, offloading, and hybrid optimizer strategies. Furthermore, the possibility of operator scheduling remains unexplored in LLM heterogenous training, thus leaving room for significant performance improvements.

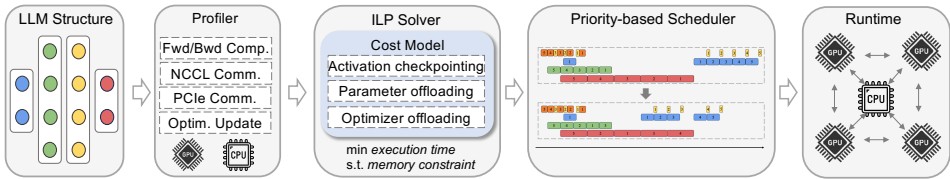

Figure 1: The overview of AutoHete.

## 3 AUTOHETE

Fig. 1 shows the overview of AutoHete. Given the LLM and hardware configuration, we construct a Profiler module to capture the memory consumption and execution time of all operators. The collected information is then fed into the Solver module, which formulates the heterogeneous training optimization problem as integer linear programming to derive optimal strategies for activation checkpointing, parameter offloading, and optimizer offloading. A priority-based scheduler further orchestrates GPU computation, CPU-GPU communication, and CPU optimizer updates to maximize overlap among these operations. Finally, the resulting execution strategy is configured into

the runtime to enable efficient heterogeneous training. Notably, our heterogeneous training system strictly maintains the data dependency of all operators so without affecting training convergence.

### 3.1 COMPUTATION AND MEMORY PROFILER

To guide the exploration of heterogeneous training strategies, we first need to collect the execution time and memory footprints of all operators. A computational graph is constructed by the *torch.fx* (Reed et al., 2022) module, which utilizes symbolic execution to capture operations invoked on a given input throughout the program. Based on the symbolic tracing mechanism, we use fake input tensors containing solely meta-data without actual values to infer the output shape of each node within the computational graph, consequently revealing the computational demands and memory allocation of each operator.

Instead of focusing on individual nodes within the computational graph, we simplify the decision-making process by considering the entire transformer block as the fundamental planning unit. Within a transformer block, there are three binary switches to consider whether to delete and recompute all intermediate activations, offload all parameters, and offload all corresponding optimizer states. This coarse-grained decision-making provides three main advantages. Firstly, the policy search space is significantly reduced, as a transformer block encompasses hundreds of nodes. Secondly, the CPU-GPU data transfer operations that aggregate multiple tensors make more efficient utilization of PCIe bandwidth. Thirdly, the coarse-grained execution time evaluation is obviously more accurate than the evaluation on individual nodes. Therefore, we partition the computational graph, where each partition represents a transformer block. The embedding layer is treated as a distinct partition.

We use $M_a$ and $M_a^{'}$ to denote the total activations and input activations of a transformer block, respectively. Note that nodes involved in in-place operations are skipped, as they are not allocated new memory footprint for their output activations. The number of parameters $M_p$ within the partition can be directly derived from the hidden dimension. The forward and backward computation overheads ($t_{fp}$ and $t_{bp}$) of the partition are evaluated by directly executing the corresponding operators on the GPU. Rather than executing on the original model, evaluation is conducted by a tiny model containing only one transformer block with identical configuration. Similar profiling is employed to evaluate the cost of CPU optimizer updates $t_{optim}^{cpu}$, GPU optimizer updates $t_{optim}^{gpu}$, parameters prefetching $t_{h2d}$, and parameter or gradients offloading $t_{d2h}$ for the transformer block.

### 3.2 SOLVER

#### 3.2.1 ANALYSIS

Before solving the heterogeneous training strategy, we analyze the GPU memory savings and the extra overheads incurred by diverse planning. For an LLM with $L$ transformer blocks, $F_i$ and $B_i$ represent the forward and backward execution of the $i$-th ($i \in \{1, ..., L\}$) transformer block, respectively. The strategy space contains three binary decision sequences: $C, P, O \in \{0, 1\}^L$, where $C[i]$, $P[i]$, and $O[i]$ represent whether to employ activation checkpointing, parameters offloading, and optimizer states offloading, respectively, for the $i$-th transformer block.

If checkpointing is applied to the $i$-th transformer block, it will reduce $2 * (M_a - M_a^{'})$ bytes GPU memory usage during the execution interval $[F_{i+1}, ..., F_L, B_L, ..., B_{i+1}]$, concurrently incurring an additional recomputation cost of $t_{fp}$. Similarly, offloading parameters of the $i$-th transformer block frees GPU memory footprint by $2 * M_p$ bytes[1] and adds extra CPU-GPU communication overhead of $t_{d2h} + t_{h2d}$ within the same execution interval.

**Insight 1:** *For activation checkpointing and parameter offloading, prioritizing earlier transformer blocks is advantageous, as it enables GPU memory savings over a longer execution duration while incurring similar costs to later blocks. This observation can be formalized as $C[i] \geq C[j]$ and $P[i] \geq P[j]$, $\forall i, j \in \{1, ..., L\}, i < j$.*

Placing the optimizer states of the $i$-th transformer block on the CPU involves 1) prefetching FP16 parameters before $F_i$, 2) offloading FP16 gradients after $B_i$, and 3) updating CPU FP32 parameters.

---

[1]The multiplication factor of 2 stems from storing parameters and activations in FP16 format during forward and backward computation.

It reduces GPU memory footprint by $12 * M_p$ bytes during the entire training iteration with a total cost of $t_{h2d} + t_{d2h} + t_{optim}^{cpu}$. Consistent with ZeRO-Offload, we avoid moving the optimizer states between the GPU and CPU since it will introduce excessive communication costs, roughly $6 * (t_{h2d} + t_{d2h})$. Despite the equivalent cost, GPU memory savings, and the duration of memory savings, disparities exist in optimizer offloading between various transformer blocks. Considering offloading optimizer states for the $i$-th block, its FP16 parameter prefetching can be overlapped with earlier forward computations $[F_1, ..., F_{i-1}]$, while its FP16 gradient offloading and CPU parameter updates can be overlapped with later backward computations $[B_{i-1}, ..., B_1]$. Appendix 6.1 provides an example of the memory allocation process.

**Insight 2:** *For optimizer offloading, the later transformer blocks should be prioritized to improve overlaps. We formulate it as $O[i] \leq O[j], \forall i, j \in \{1, ..., L\}, i < j$.*

Based on the above analysis, the heterogeneous training strategy space can be reduced to three decision variables: $\hat{c}, \hat{p}, \hat{o} \in \{0, ..., L\}$, which represent the number of sequential transformer blocks for applying activation checkpointing, parameters offloading, and optimizer states offloading, respectively. We further evaluate the peak GPU memory and total execution time of a training iteration.

### 3.2.2 COST MODEL AND STRATEGY SEARCH

**Modeling peak memory footprint.** GPU memory usage increases during the forward pass due to activation generation and parameter prefetching. Conversely, memory footprint decreases during the backward pass as activations, parameters, and gradients are deallocated. Peak GPU memory allocation occurs at the transition between these two passes. We also employed a trick to optimize memory efficiency[2]. Overall, given the GPU memory capacity $M_{gpu}$, we have:

$$2M_a' * \hat{c} + 2M_a * (L - \hat{c} + 1) + 2M_p * (L - \hat{p} + 1) + 12M_p * (L - \hat{o}) + M_{gc} \leq M_{gpu}, \quad (1)$$

where $M_{gc}$ is constant GPU storage requirements (e.g., activations) outside the transformer model.

The CPU memory is used to store offloaded model data, including optimizer states (FP32 momentum, FP32 variance, and FP32 parameters), FP16 parameters, and FP16 gradients. Notably, FP16 parameters and FP16 gradients of a transformer block can share the same memory space since they do not coexist. We define $M_{cc}$ as the CPU storage requirements beyond the transformer model (e.g., the embedding layer). Thus, given the CPU memory limitation $M_{cpu}$, we have:

$$14M_p * \hat{o} + M_{cc} \leq M_{cpu}. \quad (2)$$

**Modeling execution time.** In the forward phase, GPU computations and CPU-GPU communication for FP16 parameters can proceed asynchronously via CUDA Stream, with computation and prefetch streams overlapping to enhance efficiency. Synchronization operations are inserted into the computation stream to ensure forward computations utilize up-to-date FP16 parameters. FP16 parameters can be released immediately after computations are completed, without data transfer, as the CPU retains a copy. Thus, the critical path for forward execution is either parameter prefetching or the GPU computation stream, represented as:

$$T_{fwd} = \max \left( t_{fp} * L, \ t_{h2d} * \hat{o} + t_{fp} \right). \quad (3)$$

In the backward phase, operations such as gradients computation, activations recomputation, parameters prefetching, gradients offloading, and optimizer updates are conducted. Unlike the forward pass, the early backward pass is located at the peak duration of GPU memory usage. Premature parameter prefetching in this context carries the risk of out of GPU memory. Therefore, we consider a relatively conservative execution, prefetching parameters only one transformer block in advance. There are two cases for evaluating the synchronization overhead in the backward computation stream, i.e., whether parameter prefetching overlaps also with recomputation. We introduce $\hat{v}$ to represent the number of transformer blocks applying both activation checkpointing and parameter offloading, which can be derived by $\hat{c}, \hat{p},$ and $\hat{o}$. The synchronization overhead is denoted as $t_{sync}$, then:

$$t_{sync} = \hat{v} * \max \left( t_{h2d} - t_{fp} - t_{bp}, \ 0 \right) + (\hat{p} - \hat{v}) * \max \left( t_{h2d} - t_{bp}, \ 0 \right). \quad (4)$$

---

[2]If the optimizer states of the $i$-th transformer block reside on the GPU, its FP16 parameters are not allocated until $F_i$ or $B_i$ execution, temporarily converted through FP32 parameters, and immediately released after computation. The overhead of precision conversion on the GPU is negligible. In this context, offloading planning for FP16 parameters is unnecessary, which can be formulated as $\hat{p} \leq \hat{o}$.

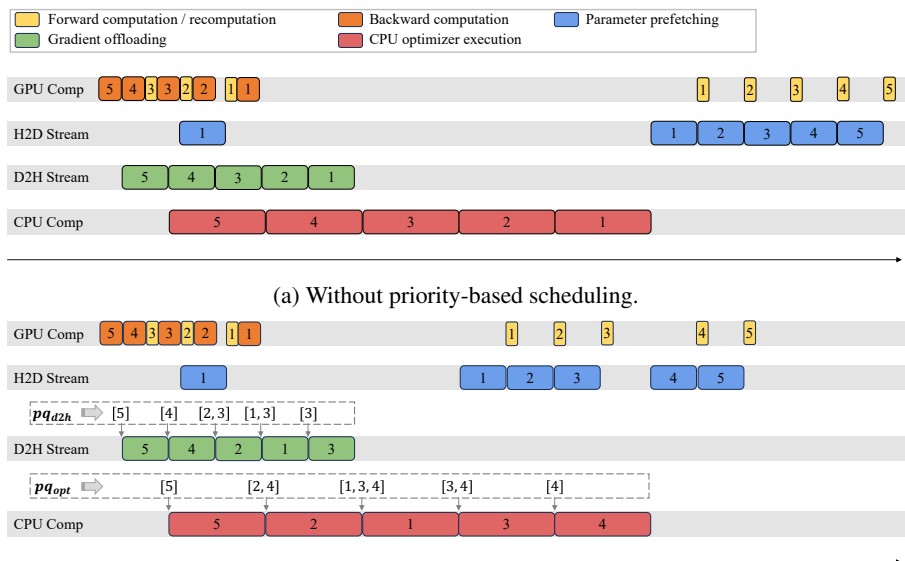

(a) Without priority-based scheduling.

(b) With priority-based scheduling.

Figure 2: Execution examples of AutoHete for an LLM comprising 5 transformer blocks, with and without priority-based scheduling strategy. Here, $\hat{c} = 3$, $\hat{p} = 1$, and $\hat{o} = 5$, indicating blocks 1-3 adopts activation checkpointing, block 1 offloads parameters, and blocks 1-5 offload optimizer states. $pq_{d2h}$ and $pq_{opt}$ denote priority queues that manage the order of blocks for gradient offloading and CPU optimizer updates, respectively. The dynamic queue states are shown in the dashed boxes.

Gradient offloading and optimizer execution are immediate, occurring as soon as gradients are generated or offloaded, without waiting for subsequent backward execution[3]. The critical path during the backward phase lies in either the CPU workflow or the backward computation stream that may be delayed by parameter prefetching. Therefore, we have:

$$T_{bwd} = \max\left( t_{bp} * L + t_{fp} * \hat{c} + t_{optim}^{gpu} * (L - \hat{o}) + t_{sync}, \quad t_{bp} + t_{d2h} + t_{optim}^{cpu} * \hat{o} \right). \quad (5)$$

**Strategy search.** With the heterogeneous memory space constraints, finding the optimal execution strategy $s = (\hat{c}, \hat{p}, \hat{o})$ becomes an integer linear programming (ILP) problem:

$$\min_{s} \ T_{fwd} + T_{bwd}, \ s.t.(1),(2). \quad (6)$$

This ILP problem can be swiftly solved by commodity solvers, as it involves only three integer variables. Given a feasible solution from ILP, we fine-tune the parameter prefetching in the backward phase to reduce the synchronization overhead. A simulator of heterogeneous training is constructed to capture dynamic GPU memory allocation. We will advance a parameter prefetching operation if enough GPU memory is available before it.

### 3.3 PRIORITY-BASED SCHEDULER

While ILP offers efficient solutions for single training iterations, there remains potential to enhance efficiency by overlapping operations across iterations. Fig. 2 illustrates the execution of AutoHete, spanning from the backward phase of the previous training iteration to the forward phase of the subsequent iteration. As shown in Fig. 2a, there are notable discrepancies in the execution speeds across streams within a training iteration: the GPU computation finishes the fastest, followed by the GPU-to-CPU communication, and finally, the CPU workflow. This gap is more pronounced in larger models due to extensive offloading of optimizer states, creating a bottleneck where the next iteration's initial computation $F_1$ must wait for the CPU optimizer updates of block 1, which

---

[3]For flexibility, we define $n$ optimizers, each responsible for performing parameter updates of individual transformer blocks.

naturally finishes last. Analogously, for operators that start the forward pass earlier, their dependent CPU optimizer updates are completed later.

**Insight 3:** *By prioritizing the execution of gradient offloading and CPU optimizer updates for earlier blocks, the idle period can be significantly reduced.*

We introduce a priority-based scheduling mechanism. Higher priority is assigned to earlier blocks in the model. Specifically, we manage two dynamic priority queues ($pq_{d2h}$ and $pq_{opt}$) to determine the execution order for gradient offloading and CPU parameter updates. Along the backward pass, we track the execution status of the gradient computation and offloading for each transformer block. It is achieved by inserting CUDA Events.

Upon the backward computation of a transformer block is completed, we append its index to the priority queue $pq_{d2h}$ of gradient offloading. Instead of offloading gradients in the backward order, we perform the next by dequeue from $pq_{d2h}$ when an offloading operation concludes. If the queues are empty, we must await the completion of the latest gradient computation. Similarly, we add the indices of completed gradient offloading to the optimizer execution queue $pq_{opt}$. Rather than following the gradient offloading sequence, we continually dequeue the transformer block indices from $pq_{opt}$ to conduct CPU optimizer updates. Notable, priority-based scheduling neither diminishes nor extends the critical path of original backward pass. It overlaps the critical path of the backward and forward pass by enabling parameter prefetching and forward computation to commence earlier.

In the subsequent iteration, we schedule parameter prefetching in the forward computation order. Necessary synchronization operations are employed to ensure data dependencies. Parameter prefetching can be launched once the CPU optimizer updates are complete. Ideally, the early initiation of parameter prefetching does not augment the peak GPU memory, as both the latter span of the backward phase and the initial span of the forward phase exhibit lower GPU memory footprints. Furthermore, they demonstrate complementary GPU memory allocation trends, wherein memory utilization gradually diminishes along the backward propagation while escalating during the forward pass. To strictly avoid running out of GPU memory, we evaluate the GPU memory footprint before scheduling parameter prefetching. If numerous gradients await offloading in the queue, which results in not enough GPU memory available, we defer parameter prefetching by one step.

### 3.4 Implementation

We implemented the AutoHete system prototype based on PyTorch. AutoHete is user-friendly and designed as a wrapper without modifying the original model definition. AutoHete leverages *torch.fx* module to capture the computational graph of the model. The system implements activation recomputation by annotating each node in the graph with checkpoint information. New nodes are inserted after each transformer block, inheriting from the *torch.autograd.Function* class with customized forward and backward functions to orchestrate parameter prefetching, gradient offloading, and optimizer updates. To maintain data dependencies, *torch.cuda.Event* is used to record and synchronize operations. Finally, the modified computational graph is recompiled to enforce the planned optimizations at runtime.

## 4 Evaluation

### 4.1 Evaluation Methodology

**Testbed.** We evaluate AutoHete on a server configured with 8 NVIDIA 40GB A100 GPU, AMD EPYC 7713 64-Core CPU, and 1TB DDR4 RAM. The GPUs are connected with NVLink, and data transfer between the GPU and CPU occurs over the PCIe 4.0 interface.

**Workloads.** Consistent with prior research, we utilized GPT-like models for performance evaluation. Models with varying scales are obtained by adjusting the hidden dimension and the number of transformer blocks, as shown in Table 1. The sequence length is 1024 for all cases.

**Baselines.** We compare the effectiveness of AutoHete with three advanced heterogeneous training solutions for LLMs. ZeRO-Offload (Ren et al., 2021) statically keeps all parameters on the GPU but offloads gradients and optimizers to the CPU. PatricStar[4] (Fang et al., 2022; Li et al., 2023)

---

[4]PatricStar has been optimized and integrated into the open-source framework ColossalAI.

Table 1: Model configuration in evaluation.

| # Params (billion) | # Layers | Hidden Size | # Params (billion) | # Layers | Hidden Size |
|---|---|---|---|---|---|
| 2 | 42 | 2048 | 4, 6, 8 | 21, 32, 42 | 4096 |
| 10, 24, 32 | 34, 80, 108 | 5120 | 12, 14, 16, 40 | 30, 35, 40, 94 | 6144 |
| 48 | 82 | 7168 | 56, 70 | 75, 92 | 8192 |

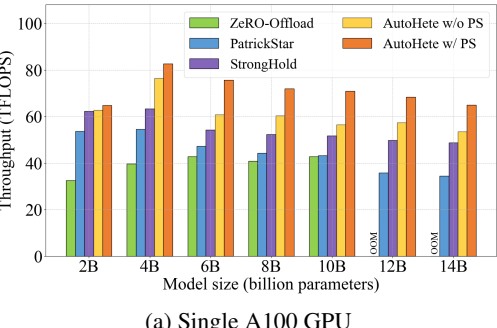

(a) Single A100 GPU

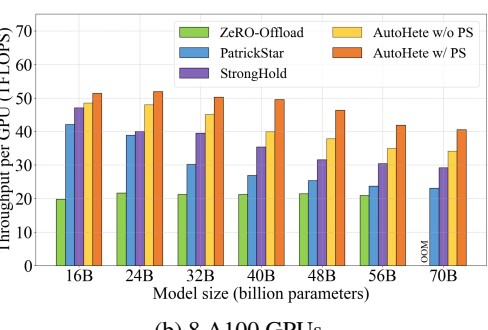

(b) 8 A100 GPUs

Figure 3: The training throughput of ZeRO-Offload, PatrickStar, StrongHold, and AutoHete across various model sizes on single and multiple GPUs.

performs on-demand placement of model data across heterogeneous memory spaces during runtime. StrongHold (Sun et al., 2022) maintains a dynamic working window on the GPU to store model data. For distributed training, we utilized the ZeRO-3 stage to partition model data.

### 4.2 EXPERIMENTAL RESULTS

We first evaluate the overall performance of AutoHete, ZeRO-Offload, PatrickStar, and StrongHold across various model scales. The performance of AutoHete without the priority-based scheduling strategy (w/o PS) is also evaluated to promote ablation studies.

**Single GPU.** Fig. 3a shows the training throughput (TFLOPS) of AutoHete, ZeRO-Offload, Patrick-Star, and StrongHold on a single GPU. Due to keeping all FP16 parameters on GPU, ZeRO-Offload runs out of GPU memory when the model size exceeds 12B parameters. AutoHete, PatrickStar, and StrongHold enable the training of larger models by further offloading FP16 parameters to the CPU.

Compared to ZeRO-Offload, we achieved 1.85x on average (up to 2.08x) performance improvement. GPU memory remained underutilized when training models with 2 billion to 8 billion parameters using ZeRO-Offload. Rather than offloading the entire optimizer states, AutoHete offloads only the requisite portion based on actual memory demands and available GPU memory, directly reducing the overheads from parameter prefetching, gradient offloading, and CPU optimizer execution. The performance gap is more pronounced for smaller models. Taking a 4B model as an example, AutoHete offloaded the optimizer states of only the last 13 out of 21 transformer blocks to the CPU.

AutoHete achieves 1.63x on average (up to 1.91x) training throughput over PatrickStar. Although PatrickStar is also memory-aware, its performance remains constrained by communication overheads and CPU workloads due to synchronous execution. Our cost model accounts for asynchronous execution between operators and focuses on maximizing overlaps. Performance gains are relatively modest for smaller models, where computational overhead dominates. The throughput improvements become more pronounced with relatively larger models. Despite considerable overheads from parameter prefetching, gradient offloading, and CPU workloads, the highly overlapped execution across the four streams effectively counteracts these costs.

StrongHold leverages data prefetching to overlap data movement with GPU computation, outperforming PatrickStar in training throughput. However, its performance remains suboptimal due to the lack of integrated consideration for the memory requirements and execution overhead associated with activations, parameters, and optimizer states. Moreover, operators overlapping in StrongHold

is limited to a single training iteration, whereas AutoHete's priority-based scheduling strategy enables overlapping across training iterations. AutoHete achieves an average of 1.32x (up to 1.41x) throughput improvement compared to StrongHold.

The priority-based scheduling (PS) mechanism contributes to a throughput improvement of 1.16x on average (up to 1.24x) for AutoHete. For 2B and 4B models, the performance of AutoHete with and without PS is comparable. This is because only the optimizer states of the initial few transformer blocks in the backward pass are offloaded to the CPU, and the introduced overhead is largely overlapped with subsequent backward computations even without PS. When more optimizers are offloaded to the CPU, CPU workflow becomes significantly behind the GPU computation stream. PS fills this gap by achieving overlaps across training iterations, i.e., scheduling parameter prefetching and forward computation of the next iteration in advance.

**Multi-GPUs.** Fig. 3b shows the training performance comparison in a multi-GPU (8 GPUs) environment. Multi-node performance results are presented in the Appendix 6.2.1. AutoHete achieves an average of 2.31x (up to 2.6x) performance improvement compared to ZeRO-Offload. Similarly, more performance gains are observed for smaller models due to reduced communication cost and CPU workload. AutoHete demonstrates performance improvement of 1.63x on average (up to 1.84x) over PatrickStar and 1.33x on average (up to 1.47x) over StrongHold. Although distributed training requires additional inter-GPU communication overheads, these overheads are effectively overlapped in AutoHete. Furthermore, the priority scheduling mechanism provides AutoHete with an average of 1.17x (up to 1.26x) performance benefit.

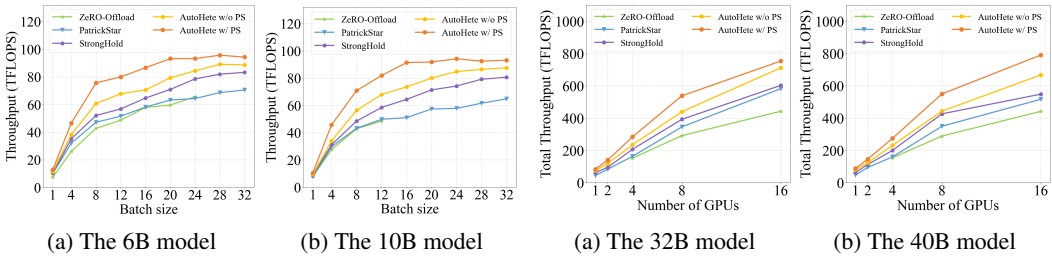

| (a) The 6B model | (b) The 10B model | (a) The 32B model | (b) The 40B model |

Figure 4: Performance on various batch sizes.    Figure 5: Performance on varying GPU scales.

### 4.2.1 FURTHER ANALYSIS

**Impact of batch size.** Fig. 4 shows the throughput variations across different batch sizes for training the 6B and 10B models on a single GPU. Larger batch sizes incur higher computational load and activation storage demands.

AutoHete initially exhibits a more rapid throughput increase, as the additional computation from larger batch sizes does not fully translate into an increase in total execution time. The backward computation overhead, including recomputation cost, is still completely hidden by communication and CPU optimizer execution. However, continuously increasing the batch size forces AutoHete to place more model data on the CPU, and the additional communication and CPU workload leads to a slowdown in throughput growth.

**Scalability.** Fig. 5 shows performance under varying numbers of GPUs for the 32B and 40B models. When scaling to more GPUs, the AutoHete system benefits not only from parallel computation but also from increased total GPU memory. This results in fewer activation checkpoints, parameter offloading, and optimizer states offloading, ultimately leading to higher throughput.

## 5 CONCLUSION

This paper presents AutoHete, an automatic and efficient heterogeneous training system for LLMs. It automatically identifies an effective execution strategy that combines activation checkpointing, parameter offloading, and optimizer offloading. A priority-based scheduling strategy is introduced to facilitate operation overlaps across training iterations. AutoHete significantly outperforms state-of-the-art heterogeneous training systems, enhancing the accessibility of LLM training.

ETHICS STATEMENT

Our work adheres to the ICLR Code of Ethics. Our research does not involve human subjects, sensitive personal data, or any identifiable information. All datasets used in our experiments are publicly available and widely adopted in prior literature. We strictly followed the terms of use of these datasets and ensured that no proprietary or private information was accessed or disclosed. The methods developed are intended purely for academic research and are not designed to produce harmful applications. We are committed to promoting fairness, transparency, and reproducibility in machine learning research, and we release our results in compliance with community standards of research integrity.

REPRODUCIBILITY STATEMENT

We provide full details to support reproducibility. The AutoHete framework, including ILP solver and priority-based scheduler, is specified in Section 3 with design assumptions. Experimental settings, evaluation metrics, and baseline configurations are described in Section 4.1. Model architectures and hardware settings are reported to allow replication of throughput measurements.

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

# 6 APPENDIX

## 6.1 MEMORY ALLOCATION ANALYSIS

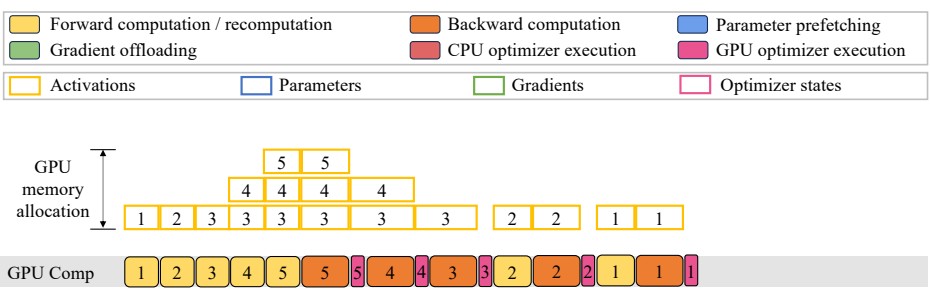

(a) The memory allocation process for activations, where only activation checkpointing is applied to the first two transformer blocks.

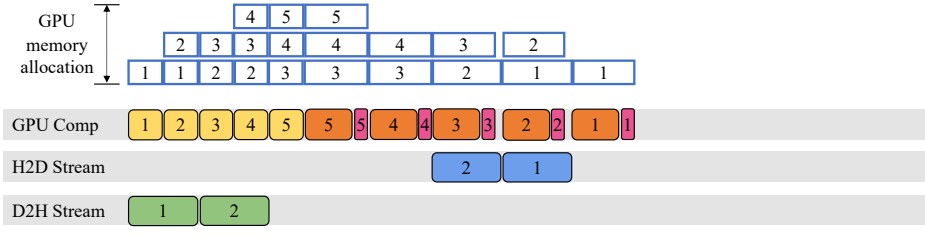

(b) The memory allocation process for parameters, where only parameter offloading is applied to the first two transformer blocks.

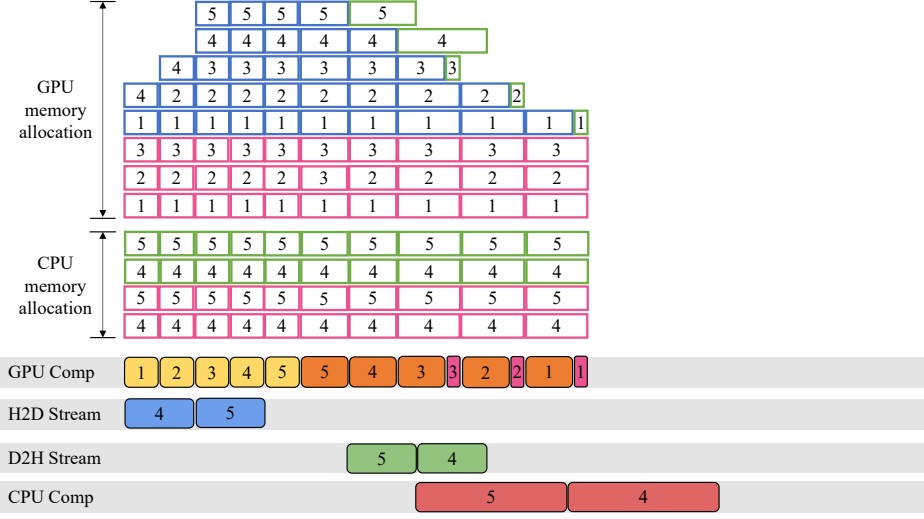

(c) The memory allocation process for parameters, gradients, and optimizer states, where only optimizer offloading is applied to the last two transformer blocks.

Figure 6: Example of the memory allocation process during a training iteration for an LLM comprising 5 transformer blocks.

Fig. 6 shows an example of the memory allocation process during a training iteration, individually applying activation checkpointing, parameter offloading, and optimizer offloading. Here, solid boxes represent executed operations, while hollow boxes indicate memory allocation along the execution flow.

Table 2: Model configuration in evaluation.

| # Params (billion) | # Layers | Hidden Size | # Params (billion) | # Layers | Hidden Size |
|---|---|---|---|---|---|
| 70 | 92 | 8192 | 80 | 84 | 9216 |
| 100, 120 | 84, 100 | 10240 | 140, 160 | 80, 92 | 12288 |

Activations are generated during the forward pass and consumed during the backward pass. As shown in Fig. 6a, applying activation checkpointing to block 1 reduces $2 * (M_a - M_a')$ bytes GPU memory usage during the execution of $[F_2, F_3, F_4, F_5, B_5, B_4, B_3, B_2]$. Similarly, for block 2, GPU memory savings occur during $[F_3, F_4, F_5, B_5, B_4, B_3]$. Activation checkpointing should be prioritized for earlier blocks. The same principle applies to parameter offloading, as observed in Fig. 6b, where earlier offloading results in a longer duration of memory savings. Due to asynchronous execution, GPU memory allocated for parameters is released only after offloading completes, while memory allocation occurs when the prefetching kernel is launched.

The memory footprint of optimizer states remains constant during execution, as they are accessed only once for optimizer updates and do not require CPU-GPU transfers. Optimizer offloading affects parameters and gradients memory allocation due to data dependencies. Blocks employing optimizer offloading prefetch the latest parameters from the CPU for forward and backward computation on the GPU, while gradients from the backward pass are offloaded to the CPU for optimizer updates. From Fig. 6c, it can be inferred that optimizer offloading should prioritize later blocks, as this allows for greater overlap of parameter prefetching, gradient offloading, and CPU optimizer updates.

GPU memory allocation for activations and parameters increases during the forward pass and decreases during the backward pass, with peak memory usage occurring at the transition between the two.

## 6.2 SUPPLEMENTAL EXPERIMENTS

### 6.2.1 EVALUATION ON MULTI-NODE ENVIRONMENT

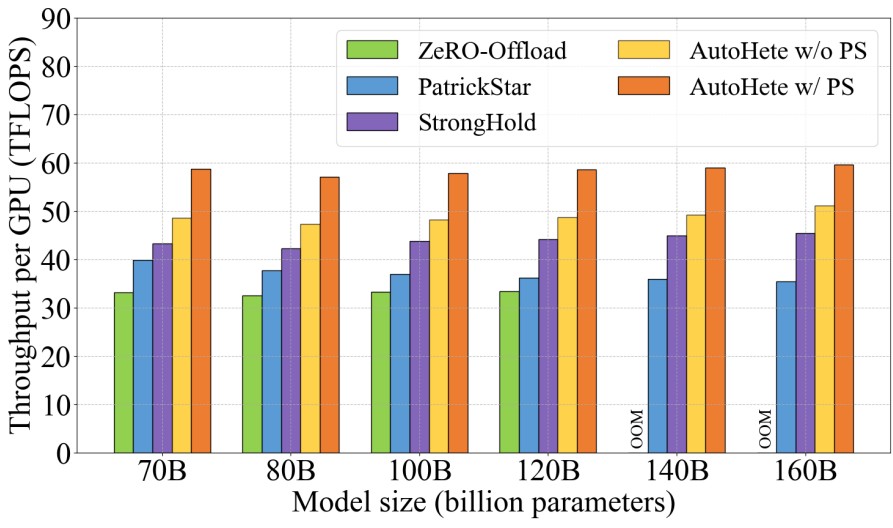

Figure 7: The training throughput of ZeRO-Offload, PatrickStar, StrongHold, and AutoHete on a multi-node environment.

We simulate a two-node cluster where each node is equipped with 8 Nvidia 40GB A100 GPUs, AMD EPYC 7713 64-core CPU, and 1TB DDR4 RAM. Inter-node communication was facilitated through InfiniBand connectivity. The model configurations are detailed in Table 2, with a global batch size of 16.

Fig. 7 shows the throughput comparison. ZeRO-Offload runs out of CPU memory rather than GPU memory when the model size exceeds 140 billion parameters, as the CPU memory capacity cannot accommodate all optimizer states and gradients. AutoHete, PatrickStar, and StrongHold address this limitation by leveraging the aggregated GPU memory to store portions of the optimizer states.

AutoHete achieves an average speedup of 1.75x (up to 1.77x) compared to ZeRO-Offload, 1.58x on average (up to 1.68x) over PatrickStar, and 1.33x on average (up to 1.36x) over StrongHold. Although cross-node distributed training introduces additional inter-GPU communication overhead, AutoHete effectively overlaps these communication costs with CPU workflow.

### 6.2.2 EVALUATION ON DIFFERENT GPU MEMORY BUDGETS

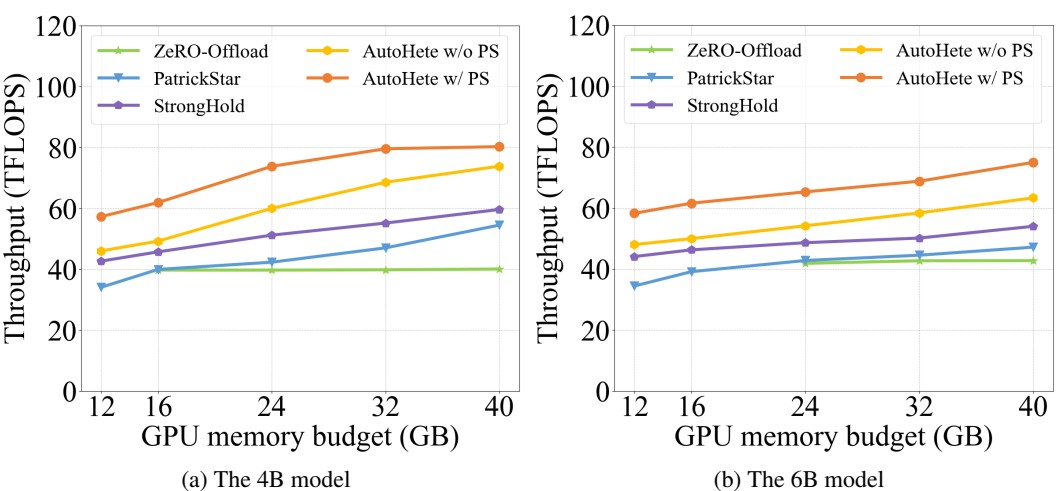

(a) The 4B model

(b) The 6B model

Figure 8: Performance on different GPU memory budgets.

Fig. 8 shows performance under varying GPU memory budgets for the 4B and 6B models on a single GPU. The throughput of AutoHete, PatrickStar, and StrongHold increases with a larger GPU memory budget, while ZeRO-Offload remains nearly constant due to its static placement strategy. Additionally, ZeRO-Offload fails to sustain 4B model training when the GPU memory budget is less than 16GB, and training the 6B model requires a minimum of 24GB of GPU memory. It is worth noting that AutoHete maintains high performance while reducing hardware costs. With merely 12GB of GPU memory consumption, AutoHete achieves higher training throughput than StrongHold, PatrickStar, and ZeRO-Offload running under the 40GB GPU memory budget.

### 6.3 THE USE OF LARGE LANGUAGE MODELS (LLMS)

We used large language models as the general-purpose assistive tool during the preparation of this paper. Its contributions were limited to improving grammar, polishing wording, and suggesting alternative phrasings for clarity and conciseness. The research ideas, methodological design, experimental implementation, analysis, and final interpretations were entirely conceived and executed by the authors.

LLMs were not used for generating novel research content, fabricating facts, or conducting scientific reasoning. All technical descriptions, results, and conclusions presented in the paper are the sole responsibility of the authors.

