# OpenReview forum: "AutoHete: An Automatic and Efficient Heterogeneous Training System for LLMs"
_ICLR.cc/2026/Conference — Submitted to ICLR 2026_

### Official Review · Reviewer_E7rK · 2025-10-25

**Soundness:** 2
**Presentation:** 2
**Contribution:** 2
**Rating:** 2
**Confidence:** 4

**Summary:**

This paper proposes AutoHete, a framework that optimizes LLM training and reduces GPU memory pressure using activation checkpointing, parameter offloading, and optimizer offloading. The problem is formulated as an integer linear program (ILP) to decide which layers should apply each of the three techniques. AutoHete comprises three components: a computation/memory profiler, an ILP solver, and a priority-based scheduler. The profiler first captures the computation graph and infers each node’s output shape using a fake input tensor. The ILP objective is built from empirical insights and a cost model that accounts for overlap between computation and communication. The priority-based scheduler favors earlier blocks for gradient offloading and CPU-side optimizer updates to reduce idle periods. With these systematic optimizations, AutoHete improves throughput by up to 1.91× on a single GPU and up to 1.84× on multiple GPUs compared to PatrickStar.

**Strengths:**

- This work targets an important, commonly encountered problem: GPU memory pressure when training large language models.

- It formulates the configuration of activation checkpointing, parameter offloading, and optimizer offloading as an integer linear program and finds solutions within a tractable search space.

- It demonstrates strong throughput improvements in both single-GPU and multi-GPU settings, and includes multi-node experiments showing effectiveness.

- The pipeline diagram and figures are clear and helpful for understanding the proposed method.

- The proposed method does not rely entirely on heuristics and is based on a derived cost model.

**Weaknesses:**

- The paper does not consider other parallelization techniques—such as tensor parallelism and pipeline parallelism—that also reduce GPU memory pressure. Experiments combining the proposed method with these techniques are needed to demonstrate effectiveness.

- The reported insights are rather trivial; developing them into formal theory with proofs would strengthen the claims.

- Numerous system-level details hurt readability. I am also unsure this paper fits ICLR well; it may be a better fit for a systems venue that values low-level optimization details.

- The flow and section structure are unclear. For example, I don’t see why Section 3.2 is titled “Solver” when it derives the cost model and formulates the problem as an integer linear program.

- The experimental setup for multi-GPU runs is unclear; the parallelism strategy used is not specified.

- The evaluation covers only a GPT-like architecture. Including recent strong models (e.g., Qwen, DeepSeek) or different architectures (e.g., MoE) would better demonstrate generalizability.

- The work relies on many fine-grained system optimizations, raising reproducibility concerns unless the authors open-source the implementation.

**Questions:**

- Parallelization coverage: Can you evaluate the proposed method in combination with tensor parallelism and pipeline parallelism to demonstrate effectiveness under common mixed-parallelism setups?

- Theoretical grounding: Can you formalize the key insights (e.g., as theorems/lemmas) and provide proofs to strengthen the claims?

- Venue fit & readability: Which machine-learning contributions make this work suitable for ICLR? Can you streamline low-level systems details or move them to an appendix to improve readability?

- Organization clarity: Why is Section 3.2 titled “Solver” if it derives the cost model and formulates the ILP? Would you clarify or restructure/retitle this section?

- Multi-GPU setup: What parallelism strategy was used in multi-GPU experiments (e.g., DP/TP/PP, ZeRO stage)? Please specify all relevant settings.

- Model coverage: Can you add results on recent strong models (e.g., Qwen, DeepSeek) and different architectures (e.g., MoE) to demonstrate generalizability?

- Reproducibility: Will you release code, configs, and scripts (including system-level optimizations) to ensure reproducibility?

**Details Of Ethics Concerns:**

This paper does not raise any specific ethical issues.

---

> ### Author Response · Authors · 2025-11-26
>
> **We sincerely thank the reviewer for the detailed feedback. We address each concern below and provide additional experimental results to strengthen our work.**
>
>
> > ### **W1 & Q1: Evaluation on other parallelization techniques.**
>
> We have integrated Tensor Parallelism (TP) into AutoHete and conducted additional experiments on 16B, 32B, 48B, and 70B models with 8 NVIDIA 40GB A100 GPUs. All speedup numbers in the table are normalized to PatrickStar as the baseline. The results show that:
> - AutoHete (TP) achieves 1.15x-1.89x speedup over PatrickStar and 1.03x-1.51x over StrongHold across all model sizes.
> - AutoHete with TP performs comparably to AutoHete with ZeRO-3, as the inter-GPU communication overhead is effectively hidden by CPU workflows.
>
> |Model size | Method |Throughput per GPU (TFLOPS) |Normalized speedup |
> |:--------:|:--------:|:--------:|:--------:|
> | **16B** | PatrickStar | 42.1  | 1.00 |
> |  | StrongHold | 47.1 |  1.12 |
> |  | AutoHete (ZeRO-3) | 51.3 | 1.22  |
> |  | **AutoHete (TP)** | **48.5** | **1.15**  |
> | **32B** | PatrickStar | 30.2  | 1.00 |
> |  | StrongHold | 39.5 |  1.31 |
> |  | AutoHete (ZeRO-3) | 50.2  | 1.66  |
> |  | **AutoHete (TP)** | **49.8** | **1.65**  |
> | **48B** | PatrickStar | 25.4  |1.00  |
> |  | StrongHold | 31.5 | 1.24  |
> |  | AutoHete (ZeRO-3) | 46.3 | 1.82  |
> |  | **AutoHete (TP)** | **47.9** | **1.89**  |
> | **70B** | PatrickStar | 23.1  | 1.00 |
> |  | StrongHold | 29.2 | 1.26  |
> |  | AutoHete (ZeRO-3) |40.6  | 1.75  |
> |  | **AutoHete (TP)** | **43.5** | **1.88**  |
>
> We emphasize that **heterogeneous training and GPU parallelism are complementary techniques.** While parallelism distributes memory across devices, it requires substantial hardware investment that is prohibitive for many researchers. Moreover, even with parallelism, the total GPU memory remains constrained. AutoHete can further extend trainable model size by leveraging CPU memory alongside distributed GPU memory.
>
> **Regarding pipeline parallelism:** PP is primarily designed for multi-node clusters where inter-node communication costs are high. Our system primarily targets resource-constrained scenarios to democratize LLM training, where memory optimization through heterogeneous training provides more immediate value. Supporting PP for larger-scale deployments is valuable future work that would extend AutoHete's applicability.
>
>
>
> > ### **W2 & Q2: Theoretical grounding and clarity of system insights.**
>
> We acknowledge that formalizing our insights would further strengthen the paper. In Appendix 6.1, we also provide detailed diagrams (Figure 6) that visualize the memory allocation and execution processes for activation checkpointing, parameter offloading, and optimizer offloading. These examples clearly illustrate how different decisions influence GPU and CPU memory usage during the forward and backward phases. In the revised version, we will further improve clarity.
>
>
> > ### **W3 & Q3: Venue Fit and readability.**
>
> We would like to clarify that AutoHete falls squarely within the official ICLR track **“infrastructure, software libraries, hardware, systems, etc.”**. This area explicitly includes research that improves the efficiency of LLM training systems.
>
> In fact, ICLR has a strong history of accepting system-oriented AI infrastructure work, including memory-efficient training and parallelization frameworks, such as:
> - Kirisame M, et al. Dynamic tensor rematerialization. (ICLR 2021)
> - Shah A, et al. Memory optimization for deep networks. (ICLR 2021)
> - Qi P, et al. Zero bubble (almost) pipeline parallelism. (ICLR 2024)
> - Sun W, et al. Co2: Efficient distributed training with full communication-computation overlap. (ICLR 2024)
> - Wang Z, et al. From promise to practice: realizing high-performance decentralized training. (ICLR 2025)
> - Liu X, et al. Netmoe: Accelerating moe training through dynamic sample placement. (ICLR 2025)
>
> AutoHete builds on this line of research by introducing system-level optimizations that significantly enhance the accessibility and scalability of LLM training. We will improve readability in the revised version by reducing unnecessary implementation details and providing clearer explanations of the insights.
>
>
> > ### **W4 & Q4: Section organization and naming.**
>
> We appreciate the reviewer’s helpful suggestion regarding the clarity of Section 3.2. The current title does not adequately reflect the scope of the section, which covers the problem analysis, the cost modeling of GPU/CPU memory and execution behavior, and the ILP-based search for the heterogeneous training strategy.
>
> We will revise the section title to more accurately represent its content—for example, “Heterogeneous Training Strategy Planning”. We will also improve the readability of this section in the revised version.

---

> > ### Author Response · Authors · 2025-11-26
> >
> > > ### **W5 & Q5: Multi-GPU setup and parallelism strategy.**
> >
> > As stated in Section 4.1, the multi-GPU experiments use the ZeRO-3 stage for distributed parallelism.
> >
> > > ### **W6 & Q6: Model coverage and generalizability.**
> >
> > We have added experiments on the Qwen3 series (4B, 8B, 14B, 32B) for both single-GPU and multi-GPU settings. The results demonstrate that AutoHete consistently outperforms ZeRO-Offload, PatrickStar, and StrongHold, achieving 1.23×–1.74× speedup. These confirm that AutoHete generalizes well to different model families beyond GPT-like architectures.
> >
> > |Number of GPUs | Model |Method |Throughput per GPU (TFLOPS) | Normalized speedup |
> > |:--------:|:--------:|:--------:|:--------:|:--------:|
> > | **1** | **Qwen3-4B** | ZeRO-Offload  | 44.7 | 0.77 |
> > |  |  | PatrickStar  | 58.2 | 1.00 |
> > |  | | StrongHold  | 48.6 | 1.18 |
> > |  | | **AutoHete**  | **87.3** | **1.50** |
> > |  | **Qwen3-8B** | ZeRO-Offload  | 42.2 | 0.83 |
> > |  |  | PatrickStar  | 50.9 | 1.00 |
> > |  | | StrongHold  | 62.4 | 1.23 |
> > |  | | **AutoHete**  | **80.1** | **1.57** |
> > |  | **Qwen3-14B** | ZeRO-Offload  | OOM | OOM |
> > |  |  | PatrickStar  | 43.9 | 1.00 |
> > |  | | StrongHold  | 58.4 | 1.33 |
> > |  | | **AutoHete**  | **76.4** | **1.74** |
> > | **8** | **Qwen3-32B** | ZeRO-Offload  | 24.8 | 0.72 |
> > |  |  | PatrickStar  | 34.4 | 1.00 |
> > |  | | StrongHold  | 45.2 | 1.31 |
> > |  | | **AutoHete**  | **55.9** | **1.62** |
> >
> >
> > > ### **W7 & Q7: Reproducibility and open-sourcing.**
> >
> > Parts of AutoHete’s implementation (e.g., parameter/optimizer offloading components and scheduling hooks) are already integrated into the open-source framework ColossalAI. We will release a complete and cleaned-up version of AutoHete upon acceptance of the paper.

---

### Official Review · Reviewer_hzBj · 2025-10-31

**Soundness:** 3
**Presentation:** 3
**Contribution:** 3
**Rating:** 8
**Confidence:** 4

**Summary:**

This work aims to make large language model (LLM) more accessible with limited GPU memory. Particularly, it introduces a training system named AutoHete, whose goal is to make use of both GPU and CPU memory by automatically deriving the configurations of three memory-saving techniques (activation checkpointing, parameter offloading, and optimizer offloading). AutoHete formulates the searching for the three techniques as an integer linear programing (ILP) problem. Additionally, it designs a priority-based scheduling mechanism that prioritizes the (off)loading of earlies layers so that the training of the next iteration would not be blocked. Evaluations show that AutoHete achieves substantial throughput improvement compared to ZeRO-Offload, PatrickStar, and StrongHold.

**Strengths:**

S1) A holistic system that jointly optimizes three memory-saving techniques (activation checkpointing, parameter offloading, and optimizer offloading) is developed.

S2) The priority-based scheduling to avoid blocking the next iteration’s training is a good solution.

S3) The empirical results are positive.

**Weaknesses:**

W1) The novelty is overall limited as all these memory-saving techniques have been investigated for long.

W2) The ILP formulation requires the model to have identical layers. When this is not met, for example, multi-modal models or hybrid (full/sparse/linear) attention models, the proposed method may not apply.

W3) The baselines are outdated (in years 2021-2022).

**Questions:**

Q1) Can AutoHete outperform mainstream LLM training frameworks, such as newly updated versions of Megatron and TorchTitan.

Q2) Please consider conducting a detailed ablation study to assess the impact of each memory-saving techniques. I believe this can help readers understand why they should be combined together.

Q3) I would also suggest doing a few case studies to show how each memory-saving technique is employed and how they are integrated.

Q4) Can AutoHete be applied to models with non-identical layers?

---

> ### Author Response · Authors · 2025-11-24
>
> **We sincerely thank the reviewer for the helpful feedback. We appreciate the recognition of our contributions and would like to address the concerns raised:**
>
> > ### **W1: Memory-saving techniques have been investigated for long.**
>
> Early memory-saving approaches targeted CNN training, where activations dominate memory consumption. These methods focus primarily on activations while ignoring model parameters. However, in LLM training, the memory bottleneck has fundamentally shifted: model data (parameters, gradients, optimizer states) now accounts for the majority of memory usage, while activations remain non-negligible. This shift renders earlier CNN-oriented approaches ineffective for LLMs.
>
> While ZeRO-Offload, PatrickStar, and StrongHold represent important advances in LLM memory optimization, they mainly focus on model data. ZeRO-Offload statically offloads all optimizer states. PatrickStar and StrongHold dynamically manage parameter placement. Both treat activation checkpointing as an all-or-nothing option applied uniformly across all layers without co-optimization.
>
> In contrast, AutoHete performs comprehensive analysis of memory footprints and execution costs across all data types (activations, parameters, gradients, optimizer states) and formulates a unified cost model. Based on hardware configuration and training requirements, our system automatically determines layer-wise strategies for activation checkpointing, parameter offloading, and optimizer offloading. This is not a simple combination but rather automated co-optimization.
>
> Furthermore, existing memory-saving systems only overlap operations within a single training iteration. AutoHete introduces a priority-based scheduling mechanism that overlaps operations across iteration boundaries. This cross-iteration scheduling represents a new optimization dimension not explored in prior LLM heterogeneous training systems and contributes up to 1.3x speedup.
>
>
>
> > ### **W2 & Q4: ILP formulation requires the model to have identical layers.**
>
> Our approach naturally handles the Transformer-based architectures used in modern LLMs (GPT, OPT, Llama, Qwen, etc.), which are built by stacking identical transformer blocks.
>
> For models with heterogeneous layers, our framework can be extended by grouping layers into coarser-grained units with similar computational and memory characteristics. The ILP formulation would then optimize over these grouped units rather than individual layers. This maintains the same optimization principles while accommodating architectural variations.
>
>
> > ### **W3: Baseline selection.**
>
> We selected ZeRO-Offload, PatrickStar, and StrongHold because they represent state-of-the-art heterogeneous training systems that are actively maintained and widely adopted. For example, PatrickStar is integrated into the ColossalAI framework and continues to be developed. ZeRO-Offload is part of the DeepSpeed ecosystem, which is extensively used in both industry and academia. StrongHold represents the latest advancement in dynamic heterogeneous training with asynchronous execution.

---

> > ### Author Response · Authors · 2025-11-24
> >
> > > ### **Q1: Comparison with mainstream LLM training frameworks.**
> >
> > Systems like Megatron primarily focus on model parallelism across GPUs to reduce per-GPU memory requirements, but they cannot overcome the aggregate GPU memory limit of the cluster and require substantial hardware investment. Our approach is orthogonal and complementary: we extend GPU memory using CPU resources, enabling training of larger models on the same hardware budget. In fact, AutoHete can be combined with parallelism techniques. Our multi-GPU experiments integrate ZeRO-3 parallelism for model data partitioning, demonstrating how heterogeneous training enhances mainstream distributed training approaches.
> >
> > Furthermore, the baselines we compare against are already integrated into mainstream LLM training frameworks. ZeRO-Offload is the heterogeneous training component of DeepSpeed, and PatrickStar is integrated into ColossalAI—both are widely used frameworks in production LLM training.
> >
> >
> > > ### **Q2: Ablation study on memory-saving techniques.**
> >
> > Our baseline comparisons actually serve as an ablation study, as the three baselines represent progressive combinations of memory-saving techniques. ZeRO-Offload uses only optimizer offloading. PatrickStar and StrongHold further add parameter offloading, extending the trainable model size. AutoHete goes one step further by integrating activation checkpointing with these offloading strategies through unified optimization.
> >
> > It is important to note that each technique alone is insufficient. Using only activation checkpointing cannot even train our smallest 2B model on a 40GB GPU, because checkpointing only addresses activation memory. Similarly, ZeRO-Offload with only optimizer offloading hits a capacity limit around 10B parameters. PatrickStar and StrongHold extend this capacity by adding parameter offloading, but their performance remains suboptimal because they do not jointly optimize with activation checkpointing.
> >
> > The consistent performance improvements we achieve (1.32x-1.91x) demonstrate that these techniques must be jointly optimized rather than independently applied. Our ILP solver systematically analyzes the memory requirements and execution costs of all three techniques together, finding the optimal combination. This holistic approach explains why AutoHete outperforms systems that only combine subsets of these techniques.
> >
> >
> > > ### **Q3: Few case studies to show how each memory-saving technique is employed.**
> >
> > We provide concrete examples to illustrate how memory-saving techniques are integrated.
> >
> > The table below presents the concrete strategies generated by the ILP Solver when training **different models** with **varying batch sizes** on a single 40GB A100 GPU. The last three columns represent the planning decisions for activation checkpointing, parameter offloading, and optimizer offloading, respectively, where the values indicate the layer ranges to which each planning is applied.
> >
> > When batch sizes are small, the recomputation overhead can be fully overlapped with partial CPU optimizer execution and CPU-GPU data transfer. In such cases, the system strategically favors more aggressive activation checkpointing. As the batch size grows, however, the solver reduces the extent of activation checkpointing and instead increases parameter and optimizer offloading to prevent GPU out-of-memory errors.
> >
> >
> > |Model size |Number of layers |Batch size | Activation plan | Parameter plan | Optimizer plan |
> > |:--------:|:--------:|:--------:|:--------:|:--------:|:--------:|
> > | 4B | 21 | 4  | 1~20  | 11~17  |  11~21  |
> > | 4B | 21 | 8  |  1~18 | 9~20  |  9~21  |
> > | 4B | 21 | 16  | 1~17  | 1~20 |  1~21  |
> > | 6B | 32 | 4  | 1~31  | 12~30  |  12~32  |
> > | 6B | 32 | 8  | 1~31  | 9~27  |  9~32  |
> > | 6B | 32 | 16  | 1~28  | 1~31  |  1~32  |
> >
> >
> > The table below presents the strategies generated by the ILP solver under **different numbers of GPUs**. As the number of GPUs increases, the system can keep more activations and model data in GPU memory, thereby reducing the need for activation checkpointing, parameter offloading, and optimizer offloading.
> >
> > |Model size | Number of layers |Number of GPUs | Activation plan | Parameter plan | Optimizer plan |
> > |:--------:|:--------:|:--------:|:--------:|:--------:|:--------:|
> > | 10B | 34 | 1  | 1~33  | 5~30  |  5~34  |
> > | 10B | 34 | 4 | 1~21  | 23~31  | 23~34 |

---

> ### Comment · Reviewer_hzBj · 2025-11-26
>
> > Early memory-saving approaches targeted CNN training, where activations dominate memory consumption. These methods focus primarily on activations while ignoring model parameters. However, in LLM training, the memory bottleneck has fundamentally shifted: model data (parameters, gradients, optimizer states) now accounts for the majority of memory usage, while activations remain non-negligible. This shift renders earlier CNN-oriented approaches ineffective for LLMs.
>
> I disagree with this statement. Activations account for a significant portion of memory consumption, especially for long-context training scenarios, and several works [1,2] have considered how to incorporate memory-saving techniques (e.g., activation checkpointing, offloading) to reduce the memory consumption of activations. Meanwhile, applying memory-saving techniques (mainly offloading) to model data has also been widely investigated in prior works like ZeRO-Offload, PatrickStar, and StrongHold.
>
> [1] Accelerating the Training of Large Language Models using Efficient Activation Rematerialization and Optimal Hybrid Parallelism. ATC 2025 \
> [2] MEMO: Fine-grained Tensor Management For Ultra-long Context LLM Training. SIGMOD 2025
>
> ---
>
> > We selected ZeRO-Offload, PatrickStar, and StrongHold because they represent state-of-the-art heterogeneous training systems that are actively maintained and widely adopted.
>
> The most deployed solutions should be Megatron, DeepSpeed, and PyTorch-native implementations (e.g., FSDP/TorchTitan). Among them, Megatron also serves as a widely used baseline in most research papers.
>
> Besides, which versions of the baselines are used in your experiments?

---

### Official Review · Reviewer_8ALu · 2025-11-01

**Soundness:** 3
**Presentation:** 2
**Contribution:** 2
**Rating:** 4
**Confidence:** 3

**Summary:**

The paper introduces AutoHete, a heterogeneous training system that dynamically integrates activation checkpointing, parameter offloading, and optimizer offloading to enhance training efficiency. The system employs a priority-based scheduling mechanism to maximize operation overlap across training iterations.

**Strengths:**

1. The paper is well-explained and easy to follow.
2. AutoHete’s performance is thoroughly evaluated across various model sizes and hardware configurations, demonstrating significant throughput improvements.

**Weaknesses:**

1. The paper does not provide sufficient validation of the accuracy of the performance model used to predict GPU memory consumption and execution time.
2. There is a lack of ablation studies to verify the effectiveness of individual components in AutoHete (e.g., the contribution of the priority-based scheduling to overall performance).
3. The framework diagrams—particularly the overview of AutoHete—are poorly designed and may hinder comprehension; they should be revised for clarity.
4. While I highly appreciate the paper’s contribution from an efficiency standpoint, its novelty appears limited, as it largely combines existing techniques with engineering-level optimizations.

**Questions:**

see weakness above.

---

> ### Author Response · Authors · 2025-11-24
>
> **We sincerely thank the reviewer for the helpful feedback. We appreciate the recognition of our contributions and would like to address the concerns raised:**
>
> > ### **W1: Validation of the accuracy of the performance model.**
>
>
> **Peak Memory Estimation:** The table below shows the comparison between estimated and actual peak GPU memory consumption across different model sizes. Our model achieves high accuracy with an average relative error of 2.6%, demonstrating that it reliably captures memory allocation patterns during heterogeneous training.
>
> |Model Size	| Estimated Peak Memory (GB) | Runtim Peak Memory (GB) | Relative Error|
> |:--------:|:--------:|:--------:|:--------:|
> | 2B |	38.2 |	38.7 |1.3% |
> | 4B |	37.8 |	38.5 |1.8% |
> | 6B |	37.6 |	38.3 | 1.8%|
> | 8B |	36.8 |	38.4 |4.2% |
> | 10B |	37.5 |	38.3 | 2.1%|
> | 12B |	37.4 |	38.6 | 3.1%|
> | 14B |	37.6 |	39.2 |4.1% |
>
>
> **Execution Throughput Estimation:** The table below compares estimated and actual training throughput. While throughput estimation shows larger variations (average 10.2% error), the model successfully identifies the optimal strategy. The discrepancy mainly stems from dynamic runtime factors like asynchronous execution and communication, which are difficult to model precisely. Importantly, our ILP solver focuses on finding the optimal configuration rather than exact throughput prediction, and the relative ordering of different strategies remains accurate.
>
> |Model Size	| Estimated Throughput (TFLOPS) | Runtim Runtim Throughput (TFLOPS) | Relative Error|
> |:--------:|:--------:|:--------:|:--------:|
> | 2B |	70.1 |	64.8 | 8.2% |
> | 4B |	89.5 |	82.6 | 8.4% |
> | 6B |	84.1 |	76.6 | 9.8% |
> | 8B |	80.8 |	72.9 |10.8% |
> | 10B |	79.2 |	71.7 | 10.5% |
> | 12B |	77.5 |	69.4 | 11.7% |
> | 14B |	73.1 |	65.2 | 12.1% |
>
> These results confirm that our cost model provides sufficient accuracy for guiding the ILP solver to find effective heterogeneous training strategies.
>
>
> > ### **W2: Ablation studies to verify the effectiveness of individual components in AutoHete.**
>
> We would like to clarify that our submission does include comparative analyses of different system components. As shown in Figure 3 of our original submission, we compared AutoHete with and without priority-based scheduling (w/o PS). The results show that priority-based scheduling provides an average speedup of 1.16× (up to 1.24×) on single GPU and 1.17× (up to 1.26×) on 8 GPUs across different model sizes.
>
> We further decompose the contributions as follows: (1) the ILP-based optimization of activation checkpointing, parameter offloading, and optimizer offloading provides the foundation for memory-efficient training, and (2) priority-based scheduling adds significant performance gains by overlapping operations across training iterations, particularly for larger models where CPU optimizer updates become a bottleneck.
>
>
> > ### **W3: The framework diagrams.**
>
> We acknowledge that Figure 1 could be improved. We will revise the diagram to better illustrate the workflow.
>
>
> > ### **W4: Combines existing techniques with engineering-level optimizations.**
>
> We clarify that AutoHete's contribution goes beyond simple combination of existing techniques. It provides systematic co-optimization through comprehensive analysis of memory allocation and execution overhead.
>
> Existing systems like ZeRO-Offload, PatrickStar, and StrongHold focus primarily on model data (parameters, gradients, optimizer states) while treating activation checkpointing as a separate, uniform option applied to all layers. In contrast, AutoHete performs comprehensive analysis of memory footprints and execution costs across all data types throughout the entire training process. We then formulate a unified cost model that captures the interactions between different memory-saving techniques. Our ILP-based approach automatically determines layer-wise strategies for activation checkpointing, parameter offloading, and optimizer offloading based on specific hardware and training requirements. This is not simply combining existing methods but rather co-optimizing them together through principled analysis.
>
> This contribution is similar to how automated hybrid parallelism systems like Alpa advanced beyond individual parallelism techniques. Just as Alpa automatically finds optimal combinations of different parallelism strategies, AutoHete automatically finds optimal combinations of memory-saving techniques through systematic modeling and optimization.
>
> Furthermore, existing systems only overlap operations within a single training iteration. AutoHete introduces a priority-based scheduling mechanism that overlaps operations across iteration boundaries. This cross-iteration scheduling represents a new optimization dimension not explored in prior LLM heterogeneous training systems and contributes up to 1.3x speedup.

---

### Official Review · Reviewer_UtWw · 2025-11-02

**Soundness:** 3
**Presentation:** 3
**Contribution:** 2
**Rating:** 4
**Confidence:** 3

**Summary:**

AutoHete formulates 3 memory saving technics: activation checkpointing, parameter offloading, and optimizer offloading. It constructs a cost model that accurately captures GPU peak memory usage and execution time. It proposed ILP to search optimal decisions for which layers to checkpoint, offload, or retain on GPU. AutoHete includes a profiler that extracts operation-level compute and memory characteristics using torch.fx. AutoHete achieves 1.32×–1.91× higher throughput than existing state-of-the-art heterogeneous training systems such as ZeRO-Offload, PatrickStar, and StrongHold

**Strengths:**

fair originality
The paper presents a novel integration of activation checkpointing, parameter offloading, and optimizer offloading into a single unified optimization framework. Each of the technics are well understood, but how to schedule them to find the best trade off remains challenging.  formulation into an ILP problem is principled approach.

good quality
The cost model can model memory/computation holistically, and transform it a well-defined optimization formulation. AutoHete is fully implemnted with requirements for model code changes. Experimental evaluation covered multiple model scales, batch sizes, GPU counts, and memory budgets. I appreicate the ablation studies

good clarify
system overview and scheduling examples are extremely helpful to illustrate operation overlap and memory allocation. The writing is effectively communicating both high-level ideas and low-level technical details, like scheduling dependencies and cost modeling.

Fair significance
AutoHete addresses the peak GPU memory contraints make training easier for researchers with limited hardware. The automatic optimization framework can speed up iteration and avoid GPU OOMs according to the hareware spec. The work represents a meaningful step toward democratizing LLM training

**Weaknesses:**

Need more originality
The three core mechanisms (activation checkpointing, parameter offloading, and optimizer offloading) have been extensively studied in prior works such as ZeRO-Offload, PatrickStar, and StrongHold. AutoHete primarily combines these methods rather than introducing a fundamentally new training technique.

The ILP-based optimization formulation, though systematic, is a straightforward formalization of the decision process (which layers to offload or checkpoint). Given the small search space (three integer variables), this formulation is conceptually simple and unlikely to be viewed as a theoretical innovation.

The priority-based scheduling strategy, while effective in practice, builds on standard ideas of operation overlap and stream prioritization used in heterogeneous training and distributed systems. Its novelty lies in application rather than mechanism.

**Questions:**

How does AutoHete’s integration of checkpointing and offloading differ fundamentally from earlier hybrid memory-management approaches? Could you argue that this integration enables qualitatively new optimization behaviors?

---

> ### Author Response · Authors · 2025-11-23
>
> **We sincerely thank the reviewer for the helpful feedback. We appreciate the recognition of our contributions and would like to address the concerns raised:**
>
> > ### **W1 & Q1: Difference between AutoHete and existing memory management methods.**
>
> Early hybrid memory management approaches targeted CNN training, where activations dominate memory consumption. These methods focus primarily on activations while ignoring model parameters. However, in LLM training, the memory bottleneck has fundamentally shifted: model data (parameters, gradients, optimizer states) now accounts for the majority of memory usage, while activations remain non-negligible. This shift renders earlier CNN-oriented approaches ineffective for LLMs.
>
> While ZeRO-Offload, PatrickStar, and StrongHold represent important advances in LLM memory optimization, they mainly focus on model data. ZeRO-Offload statically offloads all optimizer states. PatrickStar and StrongHold dynamically manage parameter placement. Both treat activation checkpointing as an all-or-nothing option applied uniformly across all layers without co-optimization.
>
> In contrast, AutoHete performs comprehensive analysis of memory footprints and execution costs across all data types (activations, parameters, gradients, optimizer states) and formulates a unified cost model. Based on hardware configuration and training requirements, our system automatically determines layer-wise strategies for activation checkpointing, parameter offloading, and optimizer offloading. This is not a simple combination but rather automated co-optimization.
>
> We believe AutoHete's contribution is analogous to the progression from individual parallelism techniques (data parallelism, model parallelism, pipeline parallelism) to automated hybrid parallelism systems (e.g., Alpa). Just as hybrid parallelism systems automatically determine the optimal combination of parallelism strategies, AutoHete automatically determines the optimal combination of memory-saving techniques.
>
>
>
> > ### **W2. ILP-based optimization formulation.**
>
> The contribution of the ILP-based solver module lies not in applying ILP as a solution method, but rather in our comprehensive analysis and formalization of heterogeneous training costs, which enables effective ILP-based optimization.
>
> We analyze how activation checkpointing, parameter offloading, and optimizer offloading interact in terms of memory savings and execution costs, and build a cost model accordingly. This cost model represents the first comprehensive formalization that jointly considers all three mechanisms for LLM training. The tables below show the estimation accuracy of our cost model. These results validate that our cost model accurately captures runtime memory and execution behavior, which is crucial for determining optimal heterogeneous training strategies.
>
> |Model Size	| Estimated Peak Memory (GB) | Runtim Peak Memory (GB) |
> |:--------:|:--------:|:--------:|
> | 2B |	38.2 |	38.7 |
> | 4B |	37.8 |	38.5 |
> | 6B |	37.6 |	38.3 |
> | 8B |	36.8 |	38.4 |
> | 10B |	37.5 |	38.3 |
> | 12B |	37.4 |	38.6 |
> | 14B |	37.6 |	39.2 |
>
>
> |Model Size	| Estimated Throughput (TFLOPS) | Runtim Runtim Throughput (TFLOPS) |
> |:--------:|:--------:|:--------:|
> | 2B |	70.1 |	64.8 |
> | 4B |	89.5 |	82.6 |
> | 6B |	84.1 |	76.6 |
> | 8B |	80.8 |	72.9 |
> | 10B |	79.2 |	71.7 |
> | 12B |	77.5 |	69.4 |
> | 14B |	73.1 |	65.2 |
>
> The small search space results from our theoretically grounded strategy space reduction. For system robustness and stability, we treat each transformer block as an atomic decision unit rather than making decisions at the granularity of individual operators. Building on this coarse-grained representation, we perform systematic analysis of how activation checkpointing, parameter offloading, and optimizer offloading affect different layers in terms of execution overhead and memory savings.
>
> The analysis reveals that these three mechanisms exhibit distinct prioritization patterns across layers, as formalized in Insights 1-2 in Section 3.2.1. Based on these insights, we further reduce the exponential strategy space ($2^{3L}$ binary decisions for $L$ layers) to just three integer variables representing the number of layers that should apply activation checkpointing, parameter offloading, and optimizer offloading, respectively. This reduction makes the problem tractable while preserving near-optimal solutions.
>
> > ### **W3: Priority-based scheduling.**
>
> Prior works typically focus on overlapping asynchronous operations within a single training iteration. Instead, our approach breaks this limitation by uniquely overlapping operations across iteration boundaries, significantly enhancing GPU and CPU utilization. This cross-iteration scheduling represents a new optimization dimension not explored in prior LLM heterogeneous training systems. Moreover, the priority-based scheduling mechanism contributes significantly to AutoHete's throughput improvement, achieving up to 1.3x performance gain.

---

### Meta-Review · Area_Chair_1BKt · 2026-01-07

**Summary:**

This paper studies an important system optimization problem in LLM training by offloading different tensors to address GPU HBM memory limitations. This is a marginal case for a difficult decision. After considering the reviewer's concern about the lack of technical depth due to the relatively natural combination of existing approaches, I have decided to reject the paper.

**Reviewer Concerns:**

The main concern is about the limited technique depth of the combination of existing offloading methods; although the authors were able to clearly explain their technique details, it does not sufficiently resolve the concern that the approach is a simple, natural combination.

**Reviewer Scores:**

I expect Reviewer 8ALu and Reviewer E7rK could potentially considering raising the score considering the additional experimental results.

---

### Decision · Program_Chairs · 2026-01-26

Reject